# Microglial Heterogeneity and Its Potential Role in Driving Phenotypic Diversity of Alzheimer’s Disease

**DOI:** 10.3390/ijms22052780

**Published:** 2021-03-09

**Authors:** Stefano Sorrentino, Roberto Ascari, Emanuela Maderna, Marcella Catania, Bernardino Ghetti, Fabrizio Tagliavini, Giorgio Giaccone, Giuseppe Di Fede

**Affiliations:** 1CNR NANOTEC—Institute of Nanotechnology, 73100 Lecce, Italy; stefano.sorrentino@nanotec.cnr.it; 2Neurology 5 and Neuropathology Unit, Fondazione IRCCS Istituto Neurologico Carlo Besta, 20133 Milan, Italy; emanuela.maderna@istituto-besta.it (E.M.); marcella.catania@istituto-besta.it (M.C.); giorgio.giaccone@istituto-besta.it (G.G.); 3Department of Economics, Management, and Statistics (DEMS)—University of Milano-Bicocca, 20126 Milan, Italy; roberto.ascari@unimib.it; 4Department of Pathology and Laboratory Medicine, Indiana University, Indianapolis, IN 46202, USA; bghetti@iupui.edu; 5Scientific Directorate, Fondazione IRCCS Istituto Neurologico Carlo Besta, 20133 Milan, Italy; fabrizio.tagliavini@istituto-besta.it

**Keywords:** microglia, neuroinflammation, Alzheimer’s disease, heterogeneity, Aβ, dementia, chemokines, MMPs, innate immunity factors, cytokines

## Abstract

Alzheimer’s disease (AD) is increasingly recognized as a highly heterogeneous disorder occurring under distinct clinical and neuropathological phenotypes. Despite the molecular determinants of such variability not being well defined yet, microglial cells may play a key role in this process by releasing distinct pro- and/or anti-inflammatory cytokines, potentially affecting the expression of the disease. We carried out a neuropathological and biochemical analysis on a series of AD brain samples, gathering evidence about the heterogeneous involvement of microglia in AD. The neuropathological studies showed differences concerning morphology, density and distribution of microglial cells among AD brains. Biochemical investigations showed increased brain levels of IL-4, IL-6, IL-13, CCL17, MMP-7 and CXCL13 in AD in comparison with control subjects. The molecular profiling achieved by measuring the brain levels of 25 inflammatory factors known to be involved in neuroinflammation allowed a stratification of the AD patients in three distinct “neuroinflammatory clusters”. These findings strengthen the relevance of neuroinflammation in AD pathogenesis suggesting, in particular, that the differential involvement of neuroinflammatory molecules released by microglial cells during the development of the disease may contribute to modulate the characteristics and the severity of the neuropathological changes, driving—at least in part—the AD phenotypic diversity.

## 1. Introduction

Alzheimer’s disease (AD) is a neurodegenerative disorder marked by neurotoxic protein aggregates, composed mainly by amyloid-beta (Aβ) peptides and hyper-phosphorylated tau protein (p-tau) accumulating within the brain. The formation of these toxic assemblies begins years—probably even decades—before the clinical onset of the disease [1] and is associated with additional and yet unexplored molecular mechanisms which drive the deleterious effects of the two main misfolding peptides (i.e., Aβ and tau) [2]. The imbalance between the overproduction of misfolded proteins and the malfunction/ineffectiveness of the endogenous clearance systems results in the formation of toxic aggregates which, as a consequence, trigger a neuroinflammatory reaction [3,4,5]. The term neuroinflammation refers to a physiological response occurring within the nervous system against a neuronal injury that, in case of AD, is most likely a consequence of the neurodegenerative process sustaining the disease. The inflammatory response is promoted by cellular players (glial, endothelial, and immune cells) and molecular released factors (cytokines, chemokines, reactive oxygen species, and secondary messengers) generating a complex microenvironment that can modulate the effects of the two main misfolding proteins of AD. Neuroinflammation is therefore an early event that occurs since the asymptomatic stage of the disease [6,7,8] and, relying on the patient-specific response, contributes to the development of the disease in a highly heterogeneous manner [9]. Moreover, the propensity to develop a chronic, sterile, low-grade inflammation increases with ageing and has been called “inflammaging”. It may substantially contribute to the pathogenesis of age-related diseases such as AD and is sustained by a variety of stimuli, including endogenous cell debris and misplaced molecules [10].

Increasing evidence from recent scientific literature strongly supports the view of AD as a highly heterogeneous neurodegenerative disorder occurring under several distinct clinical and neuropathological phenotypes [11,12]. However, the molecular bases of such variability remain largely unexplored [13,14,15,16]. More detailed knowledge of the mechanisms underlying AD heterogeneity could help to improve the diagnostic protocols for the disease and provide grounds to design tailored therapies for specific AD subgroups.

As suggested by several reports supporting the relevant association between neuroinflammation and the development of AD [17,18], the study of the neuroinflammatory events in AD could provide new insight into the pathological mechanisms of the disease and, at the same time, contribute to explain the heterogeneity showed by AD patients [19]. Microglial cells are one of the most important players in neuroinflammation [20,21]. They are branched-shape cells considered as resident phagocytes of the central nervous system (CNS) ranging from the 0.5–16.6% of all human brain cells. Morphologically and functionally changeable, these cells constantly surveil the surrounding neuronal environment, extending their motile processes to protect nervous tissue from any possible threats [22]. When activated, their branches become shorter, their soma hypertrophic, and microglia acquire the competence to phagocyte and digest molecular targets. At the same time, microglia trigger the immune response by releasing several pro-inflammatory cytokines, such as interleukin-1β (IL-1β), IL-6, IL-12, IL-18, interferon-gamma (IFN-γ), tumor necrosis factor-α (TNFα) [23,24,25,26,27]; chemokines (i.e., CCL2, CCL5, CXCL9) [28]; numerous endogenous proteolytic enzymes like matrix-metalloproteinases (MMPs) [29]; and factors related to the innate immunity, such as Lipocalin-2 (LCN2) and CD14 [30,31]. Similarly, by secreting anti-inflammatory cytokines like IL-1rn, IL-4, and IL-10 [32], microglial cells are able to auto-regulate their actions, leading the neuroinflammatory process to resolution. In AD, microglia can bind Aβ peptides, triggering a massive neuroinflammatory reaction aimed at avoiding the accumulation of Aβ in the brain [33]. However, the continuous overproduction of the Aβ constantly reactivates the inflammatory processes that, over time, become detrimental. This process develops differently among patients through the secretion of specific pro- and anti-inflammatory factors that may modulate the severity of pathology contributing to its variability too.

The objective of this study was to investigate the potential contribution of microglia to AD phenotypic variability. To this end, we carried out a neuropathological and biochemical study on a series of AD brain samples, providing evidence about the heterogeneity of microglia in our cohort of AD patients. We found that such heterogeneity is associated with the secretion of distinct pools of pro- and/or anti-inflammatory molecules which most probably play an active role in modulating the onset, severity and progression of the disease. These data offer additional insight into the relationship between AD pathology and microglia-mediated neuroinflammatory response and further suggest a potential role of the microglial cells—and their released molecules—in AD phenotypic diversity.

## 2. Results

### 2.1. Morphological/Functional Profiling of Microglia in AD Brain Samples

The morphological/functional profile of microglial cells was analyzed by immunohistochemistry in a well-defined cohort of 24 AD patients previously characterized in our lab [14] (Appendix A). Fifteen out of 24 cases were available for the immunohistochemical study of microglia. Frontal cortex samples were immunostained with the anti-IBA1 antibody to dissect morphology and distribution of both the quiescent and the activated forms of microglia. Results demonstrated significant dissimilarities concerning density, shape, and distribution of microglial cells among both familiar (fAD) and sporadic (sAD) cases (Figure 1 and Figure 2).

Specifically, the APPA673V homozygous carrier (fAD1), previously characterized by abundant amyloid deposits both in the parenchyma and in the vessels [34], showed a particular distribution of microglia around vessels and amyloid plaques (Figure 1a,d,g). Microglial cells appeared in their amoeboid state [35], with hypertrophic soma and completely disappeared branches (Figure 1j). These features are usually associated with the phagocytic state of microglia [36,37]. Quantitative analysis of the immunostaining in fAD1 (Figure 1m) indicated strong immunoreactivity for IBA1 and a high cellular density compared to the other AD patients. Amoeboid microglia were recognizable also in the APPA713T (fAD2) case (Figure 1b,e,h) that was reported to have severe Cerebral amyloid angiopathy (CAA) and low-density parenchymal amyloid deposits in the neuropil [38]. However, although fAD1 and fAD2 cases share the same morphological features, microglia observed in fAD2 was not organized in a specific pattern and was homogeneously distributed in all cortical layers. Patients carrying mutations in presenilin genes showed distinct characteristics (Figure 1c,f,i). Despite the young age of death (age 43), microglia of PS1P117A carrier (fAD3) resulted dystrophic Figure 1l), a condition usually associated with normal ageing [39]. The distribution was homogeneous and did not present with any specific organization. However, as shown by densitometric analysis, fAD3 patient displayed the highest overexpression of IBA1 protein and the greatest density of microglia among all the AD cases tested in this study (Figure 1o).

Heterogeneity of microglia was also consistently observed among sAD patients.

Figure 2 shows cases sAD2, sAD15, and sAD17. The first two patients showed strong microglial activation as underlined by quantification of the IBA1 immunoreactivity, the densitometry, and the morphological analysis. sAD2 showed a specific distribution of microglia in a bilayer pattern (panel a) with higher density in the upper and lower layers of the cortex. Both the number and intensity of positive pixels, as well as the morphology of microglia, recalled those observed in the fAD1 patient, even if cells appeared more roundish and ramified without any particular organization. sAD15 exhibited a rod-shaped morphology microglia (panel k) characterized by an enlarged cell body with few, thin branches arising almost perpendicularly from the rod-shaped cytoplasm. Microglia were homogeneously distributed, but compared to the fAD3 case, the density of positive pixel/μm^2^ was 25% lower. sAD17 showed an inferior percentage in both the number and intensity of IBA1 positive pixels, expressing a more limited microglial activity compared to the other cases (panel o). Morphologically, microglia appear dystrophic, with beaded, fragmenting processes and spheroids comparable to fAD3 but to a lesser degree. Other patients showed characteristics similar to those remarked upon in the cases reported below (Appendix A). Specifically, sAD8 displayed activated microglia that was mainly organized around vessels. (Appendix A) Similarly to the sAD17 patient, sAD13, sAD18, sAD7, and fAD4 (Appendix A) had more ramified microglia, with the last two cases showing focal areas of intense activation. We also observed rod-shaped microglia in sAD12 patient (Appendix A)—like in sAD15 brain sample—and roundish cells in sAD3, sAD6, and sAD9 cases (Appendix A) comparable to the sAD2 patient but with slight intensity in IBA1 signal.

The low number of cases available for this study did not allow a stratification of patients based on specific microglial profiles. However, the overall observations in the immunohistochemical study suggested a consistent heterogeneous picture of microglia in AD with differences regarding the cellular morphology, density, and distribution, as well as resting/activation state of microglial cells, in both genetic and sporadic AD cases.

### 2.2. Neuroinflammatory Cytokines in AD Patients

The results of the neuropathological study on microglia further supported previously reported evidence on the variability of the functional state of microglia in AD [40,41]. Following this view, we analyzed the microglia released molecules to check whether this variation could be associated with differences in the production and release of inflammatory cytokines. Thus, brain homogenates (AD *n* = 24; CTRLs *n* = 6) from frontal cortex were analyzed by multiplex assay to test a panel of pro- and anti-inflammatory factors (Appendix A), known to be expressed by microglial cells and reported by the scientific literature as molecules potentially involved in AD pathogenesis [7]. The analysis showed a higher expression of the overall measured analytes in AD samples respect to controls, indicating that, regardless of the potential pro- or anti-inflammatory effect, neuroinflammation is exacerbated in AD patients at variance with non-demented controls (Appendix A). Specifically, IL6 resulted in 2.5-fold increase in comparison with the median of controls, followed by CCL2, CCL5, CXCL9, CXCL10, MMP-7, and MMP-8 whose levels were all 1.5-fold increased respect to control samples (Appendix A).

Statistical analysis showed significantly higher concentrations of IL-4 (*p* = 0.0357), IL-13 (*p* = 0.0050), IL-6 (*p* = 0.0183), CCL17 (*p* = 0.0447), CXCL13 (*p* = 0.0008), and MMP-7 (*p* = 0.0041) in AD samples (Table 1 and Figure 3). Interestingly, CXCL13, the most significant factor detected in brain homogenates, has never been directly associated with AD, to our knowledge.

Lastly, we considered the dichotomization in pro (i.e., IFN-γ, IL-1ɑ, IL-2, IL-6, IL-12 p70, IL-18) and anti-inflammatory (i.e., IL-1rn, IL-4, IL-13) cytokines, measuring their levels in both control and patient groups. Interestingly, we observed a predominance of anti-inflammatory cytokines in both groups with marked statistical significance in AD patients (*p*-value < 0.0001) (Figure 4).

All recovery for each calibrator fell within 80–120% of the known value (data not shown). However, since IL-10, IL-1β, IL-6, IFN-γ, IL-12, CCL17, and CXCL10 were close or below the lowest calibrator values—and so not in the quantitative range of the assay for most of the samples analyzed—we think they should be interpreted carefully. Surprisingly, TNFα, IL-10, and IL-1β, key players of the inflammatory reaction that were related to the central and peripheral immunological response to AD [42,43], were undetectable in more than 50% of samples of our cohort of brain homogenates and therefore excluded from the analysis.

### 2.3. Correlations between Inflammatory Molecules and Clinical, Neuropathological and Biochemical Features of AD Cases

We used STRING webtool to generate a Protein-Protein Interaction (PPI) network to determine the direct and indirect connections among the inflammatory molecules analyzed in this study (Figure 5a). The PPI network showed three main groups (i.e., group 1: IL-6, IL-12a, IL-12b, IL-4, IL-13, IL-18, IL-1a, IL-1rn; group 2: CX3CL1, CXCL9, CXCL10, CXCL13, CCL5; group 3: MMP-1, MMP-7, MMP-8, MMP-9, LCN2). Supported by STRING analysis, we grouped the neuroinflammatory factors in these 3 classes to reduce the number of variables and simplify the correlations with clinical and biological features. We moreover restrained LCN2 and CD14 in a fourth group (Innate Immunity Factors, IIF) considering the high biological distance and slight connection link (co-expression) to MMPs. To assess the contribution of each specific family of inflammatory factors (i.e., cytokines, chemokines, MMPs and IIFs to the inflammatory process associated with AD, we firstly analyzed the abundance of each family on the total amount of factor released by microglial cells. We found a prevalence of immunity factors (38%) and MMPs (37%) followed by chemokines (19%) and cytokines (5%) (Figure 5b).

We then investigated on the potential correlations between the four subgroups of inflammatory factors and (i) clinical (i.e., age at onset/death, disease duration), (ii) neuropathological (i.e., amyloid deposits dispersion index (DI)), and (iii) biochemical (i.e., soluble—Aβ40s and Aβ42s—and insoluble—Aβ40i and Aβ42i—Aβ levels in fractionated brain homogenates) disease indicators, as shown by the heatmaps in Figure 5c and Appendix A. The analysis showed that:the levels of Aβ in the fractionated brain homogenates are positively correlated with cytokines and chemokines and negatively correlated with MMPs and IIFs (Figure 5c). These data are in compliance with the hypothesis that the release of cytokines/chemokines is induced by the increasing production and accumulation of Aβ in AD brains [44,45,46] and suggest a possible protective role of LNC2 and CD14, composing the IIFs family, which are released by activated microglia and are associated with lower levels of Aβ in the brain.the DI was positively correlated with chemokines and cytokines and inversely correlated with LNC2 and CD14 (Figure 5c). So, the higher is the concentration of LNC2 and CD14 in the brain, the lower is the DI value, which implicates the presence of a lower number of amyloid deposits with a trend to form larger plaques. This result may suggest a role of the innate immunity molecules in slowing amyloidosis associated with AD.Among clinical parameters, age at onset and age at death are positively correlated with the levels of chemokines and cytokines (Figure 5c) while disease duration shows a very weak correlation with the four subgroups of inflammatory factors.the correlation between the MMPs family and all the disease indicators are globally weak (Figure 5c). This weakness results from the opposite effect of MMPs. Individually analyzed, MMP-1 and MMP-8 show a robust association, positive and negative respectively, with the age at onset/death and the Aβ42i levels (Appendix A). MMP-9 shows a negative association with Aβ40 (both soluble and insoluble), and MMP-7 shows a mild/weak positive association with all the disease indicators.

The low number of cases in our cohort represents a limitation in the interpretation of these results.

### 2.4. Patients Stratification and Cluster Composition

In a further step of our study, patients were grouped on the base of the expression of each neuroinflammatory molecular family using the Hierarchical Cluster Analysis (HCA) starting from the compositional dataset. The HCA sets yielded a dendrogram based on the Euclidean distance and the complete linkage (Figure 6a), where the higher is the value expressed in ordinate, the greater is the difference between two clusters.

On the basis of this dendrogram, our AD cohort can be divided in three clusters: AD-CL1, composed by eight patients (fAD2, sAD2, AD3, sAD6, sAD5, sAD7, sAD15, sAD16); AD-CL2, composed by seven patients (sAD9, sAD10, sAD1, fAD1, sAD20, sAD13, sAD18); AD-CL3 composed by nine patients (sAD19, sAD11, sAD14, sAD8, sAD12, fAD4, sAD4, sAD3, sAD17). Figure 6b shows the relative abundance of the four neuroinflammatory subgroups in each cluster, while Table 2 summarizes the mean per cluster for all the disease indicators analyzed in this study.

AD-CL1 was characterized by the predominance of IIFs, with high levels of CD14 (see Appendix A), a molecule involved in the microglia-mediated Aβ clearance. Accordingly, this cluster displayed the highest number and intensity of IBA1 positive pixels, which means an intense microglial activation, and the lowest DI (corresponding to the low number and larger size of Aβ deposits), suggesting an active clearance of amyloid aggregates (that may act as seed for new deposits) by microglia. AD-CL1 showed also the earliest age at onset and the earliest age at death (Table 2).

AD-CL2 patients had all the inflammatory factors elevated without a clear prevalence of a specific neuroinflammatory molecular family and the highest amount of cytokines and chemokines among all clusters. AD-CL2 was characterized by the highest levels of Aβ42 and Aβ40 species in both soluble and insoluble fractions of brain homogenates, by the highest age at onset but also by the fastest disease progression rate together with the lowest Braak stage. These features may be interpreted as a strong tendency of Aβ deposition to induce derangements in the neuropil, both as neuroinflammation and as tauopathy, likely inducing an aggressive clinical phenotype (Table 2).

AD-CL3 was characterized by a low neuroinflammatory profile (lowest number and intensity of IBA1 positive pixels; lowest levels of cytokines), probably due to the lower activation of microglial cells (Table 2). Indeed, AD-CL3 was the cluster with the lowest amount of almost all the molecules tested in this study, except for MMP-8, MMP-9, CX3CL1, and LCN2 (Appendix A). The MMP family was the most represented molecular subgroup in this cluster, which was also characterized by the low levels of Aβ peptides in the brain and the longest disease duration.

The Kruskal–Wallis test was performed to detect differences among clusters in disease indicators. Results in Table 2 suggest that data do not show any statistically significant difference among the three cohorts of AD patients aside from the slight significance of the Aβ42s levels (*p*-value 0.0347). However, the Kruskal–Wallis test, as a nonparametric test, is characterized by a lower power than its parametric (distribution based) counterpart, and it may result not powerful enough to detect real differences when the sample size is small.

## 3. Discussion

Neuroinflammation is a fascinating and still largely unexplored aspect of AD and it is involved either in pathogenesis, development, and—at least in part—in the generation of different phenotypes [47]. Microglia is the crucial cellular player of inflammation in CNS. Microglial cells can exist in different functional states [32,48], which are recognizable on the basis of the different proteins produced and released in the extracellular environment. These proteins, encompassing cytokines, chemokines, metalloproteinases, and other immunity factors, mirror the brain neuroinflammatory condition and could be ultimately used to stratify patients in different subgroups characterized by distinct ‘neuroinflammatory phenotypes’ for more tailored therapeutic studies [49]. However, the extreme variability of the activation of microglia among AD brains and even within the same brain area suggests a more complex and unmapped involvement of neuroinflammatory players (cells and cytokines) in the pathogenesis of AD and its phenotypic variability. Following this view, this study was aimed at the characterization of microglial cells and their associated inflammatory molecules in the frontal cortex of both sporadic and familial AD patients, and at investigating their possible implication in the occurrence of the disease under distinct phenotypes.

We documented that microglial cells were differently represented within the brain of both familiar and sporadic forms of AD regarding morphology, distribution (spatial organization), density and intensity of the activation. Morphologically, in some patients, microglial cells appeared in their amoeboid state, with hypertrophic soma and complete disappearance of branches, typical features associated with the phagocytic state of microglia; in other patients, they resulted dystrophic with beaded, fragmented processes and spheroids, while in other ones were rod-shaped with enlarged cell bodies and branches reduced in number and thickness. It is known that in AD brains microglia show principally two distinct morphological phenotypes: the “reactive”/amoeboid-like M1 phenotype and the homeostatic-like morphology (M2 phenotype) [50]. Our work suggests that microglia undergo a more complex morphological and functional remodelling in AD brains, and reinforces the view that the dichotomized morphology of microglial cells is an over-simplified interpretation, unable to cover the variability in spatial organization, density and shape of microglia observed in AD patients [5,19,51]. Such morphological differences were also highlighted by the study of El Hajj et al., from an ultrastructural point of view [52]. By dividing hippocampal areas of APPSwe-PS1Δe9 mice into subregions, according to pathological insult, the authors identified distinct microglial ultrastructural aberrations related to AD pathology. Importantly, several of these ultrastructural alterations were also observed in the human hippocampus of AD patients. In particular, phagocytic microglia recognized in fibrillar amyloid-β areas look like, although at pure morphological level, microglia associated plaques observed in this paper with enlarged cell bodies in the proximity of Aβ deposits. However, the underlying mechanisms that trigger changes in the morphology of microglia remain elusive so far [52]. Concerning the spatial organization, and thus the distribution within the brain tissue, we found (i) cases in which the microglia were not organized in a specific pattern but was homogeneously distributed (ii) patients in which the microglia cluster around vessels and (iii) patients in which the microglia activation was more pronounced organized in the superficial and the lower cortical layers. Regional heterogeneity of microglia has been already reported in the literature and may be a result of the resident environment, due to the interaction with local neurons and the epigenetic landscape that may alter microglia behavior [53,54]. The number of microglial cells was also highly variable from patient to patient. Indeed, we found brain samples with a very low density of microglial cells within the parenchyma and cases with a high abundance of microglia. Furthermore, microglia appeared activated in almost all AD patients—coherently with the late stage of the disease (Braak stage for almost patients: V-VI)—even if some samples showed a very low IBA-1 signal.

Recently, through the use of multiplexed mass cytometry, Böttcher and colleagues have clearly demonstrated the presence of specific microglial subsets, each one characterized by a peculiar signature, examining distinct human brain areas [55]. The existence of human microglial phenotypes in neurodegenerative diseases supports our attempt to demonstrate the importance of the microglial heterogeneity in the pathological context of AD. Indeed, each microglial subset could reasonably impact on the neuroinflammatory response differently, leading to selective vulnerability to AD pathology in distinct brain areas and being a source of phenotypic variability in AD patients.

In their activation state, microglial cells change their morphology and release a number of neuroinflammatory molecules whose implication in AD pathology has been recognized. For example, elevated levels of TNFα and IL-6 were found in the serum and brain tissue of AD patients, respectively [56,57]. Following an approach proposed by Chen and colleagues [40], consisting of a multiplex analysis of the neuroinflammatory molecules secreted by microglia, we observed that the overall levels of the microglia-derived neuroinflammatory factors were significantly higher in AD samples than in controls, regardless of their pro- or anti-inflammatory role. The increase of pro-inflammatory cytokines can be explained by the state of chronic inflammation caused by the continuous overproduction of pathogenic entities, such as Aβ aggregates, which is characteristic of AD onset and progression. Moreover, the brain samples used in this study came mostly from patients in advanced stages of the disease, in which a high neuroinflammatory state is conceivable. On the other hand, the presence of elevated anti-inflammatory factors is not surprising considering that the anti-inflammatory pathways may reflect the negative feedback signaling aimed at limiting an excessive inflammatory reaction against misfolded proteins (i.e., Aβ and tau). This view is in agreement with previous studies on animal models showing that the stimulation of pro-inflammatory signaling pathways leads to strong neuroinflammation able to lessen the amyloid burden [58,59]. In these terms, the neuroinflammatory process is not harmful in itself but becomes detrimental if the inflammation is unable to proceed to a resolution [60]. In line with this view, our data showed a significant predominance of anti-inflammatory over pro-inflammatory cytokines (i.e., IL-1rn, IL-4, IL-13) in the AD patients. This result might indicate that molecules aimed at reducing the neuroinflammation remain active even in the last stages of the disease and provide a rationale for the use of anti-inflammatory compounds for therapeutic approaches in AD [61].

In our study, we found statistically significant high levels of IL-4, IL-13, IL-6, CCL17, CXCL13, and MMP-7. Interestingly, while IL-4, IL-13, IL-6, CCL17, and MMP-7 have been widely described in association with AD [29,42,62], CXCL13—to our knowledge—has never been linked to AD. CXCL13 is selective chemotactic for B-1 and B-2 cells and elicits its effects by interacting with the CXCR5 receptor, expressed by both microglia and T- and B- lymphocytes. High CXCL13 levels in the cerebrospinal fluid were demonstrated to correlate with increased B-cell recruitment in the CNS in a number of human disorders such as lymphoma, Lyme disease and multiple sclerosis (MS) [63,64,65]. In patients affected by MS, CXCL13 is involved in the recruitment of Th1, Th17, and B-cells [66], representing a useful marker for the diagnosis [67]. Moreover, in a MS animal model, it has been demonstrated that CXCL13 and its related pathways contribute to disease pathogenesis. To date, there is no evidence supporting a role of CXCL13 in the pathogenesis of AD. However, several studies demonstrated the involvement of Th1 and Th17 cells in the AD-related neuroinflammatory process. Th17 cells have been hypothesized to be directly responsible for neuronal cell death in AD [68]. Additionally, B-cells have been seen to be involved in AD through the production of autoantibodies, produced also in response to the Aβ toxic aggregated and to oligomers and protofibrils both in the peripheral and CNS [69,70]. All these data suggest an active role of CXCL13 in neuroinflammation associated with AD that should be further investigated.

Using the STRING webtool, we subsequently grouped the inflammatory molecules based on their molecular functions, connections and interactions in four different families: cytokines, chemokines, MMPs, and IIFs, the last two being the most represented. High levels of both MMPs and IIFs have been reported in the brain of AD patients [29], where they are involved in the clearance of Aβ aggregates [71,72] and the shutdown of inflammation [73]. Despite the division in families facilitated the analysis, this process can hide individual effects, as in the case of the MMP family. Indeed, MMP-1 and MMP-8 showed opposite association indexes to the same disease indicators (Appendix A), lowering the family global values of association. To address this point, we tried to subdivide the group according to the functional classification of the MMPs (Appendix A). In this way, however, MMP-1 and MMP-8 are part of the same group (i.e., collagenases). MMP-1 is poorly represented in our cohort so, when we grouped it with MMP-8, we obtained a subgroup highly correlated with MMP-8 (i.e., Pearson’s correlation coefficient among MMP-8 and the new subgroup is equal to 0.999) (Appendix A). This means that grouping MMP-1 and MMP-8 together leads to masking the positive association between MMP-1 and disease indicators. Considering the small sample size, further investigation will be needed to clarify this point. Eventually, we used the inflammatory families to group AD patients into three clusters: AD-CL1, AD-CL2, and AD-CL3.

AD-CL1 was characterized by a predominance in the expression of IIFs, particularly CD14, and clinically by the earliest age at onset and the earliest age at death. Moreover, this cluster presented an intense microglial activation and a low DI, that means a lower number of Aβ deposits having a larger size, in comparison with the other two clusters. This strong microglial activation is accompanied by low Aβ42 levels in both the soluble and the insoluble fractions. Indeed, the expression of CD14 is needed to microglia for binding the Aβ42 fibrils and the subsequential clearance through the lysosomal pathway [74,75,76].

AD-CL2 was characterized by the fastest disease progression, suggesting a more aggressive clinical phenotype. From a neuroinflammatory point of view, AD-CL2 did not show any predominance of specific neuroinflammatory family but presented the highest expression levels of cytokines and chemokines in comparison with the other subgroups. Moreover, accordingly with its aggressive phenotype, AD-CL2 presented the highest amounts of Aβ42 and Aβ40 species. The exacerbation of the neuroinflammatory reaction is probably caused by the high amount of Aβ species (40 and 42), which are known to bind microglial surface receptors like CD36, TLR4, and TLR6, resulting in the production of pro-inflammatory cytokines and chemokines released from activated microglia [77,78]. Moreover, the age-associated microglia senescence can negatively impact the microglia functionality and eventually trigger and sustain neurodegeneration [79,80]. This strong inflammatory reaction surely impacts on the clinical outcome of the disease and could explain the short duration of the illness in this cluster.

Lastly, AD-CL3 was marked by a poor neuroinflammatory profile, which may be the cause or the consequence of the lower activation of microglial cells. Indeed, AD-CL3 is the cluster with the least amount of almost all the molecules tested except for MMP-8, MMP-9, which resulted highly expressed, followed by CX3CL1 and LCN2. MMPs have been extensively and convincingly involved in the degradation and clearance of Aβ [71,81]. This may explain the small amount of Aβ in brain samples from AD patients composing this cluster, as well as the longer disease duration, compared to AD-CL1 and AD-CL2. Interestingly, increasing levels of MMP-8 and MMP-9 have been related to cerebrovascular damage and dysfunctions [82,83]. In our study, AD-CL3 patients, who express the highest level of MMPs among the families, and specifically abundant level of MMP-8 and MMP-9, shows also the lowest extent of CAA pathology. In this context, the predominance of MMPs seems to be beneficial. However, this data is not in accordance with Lee and colleagues who reported a pro-inflammatory action of MMP-8 by modulating TNF-α activation [84]. Of note, TNF-α resulted poorly expressed in our samples. Moreover, the overexpression of CX3CL1 and LCN2 explicates a protective role, modulating the severity of the neuroinflammation process: the expression of CX3CL1 is significantly decreased in AD and inversely correlated to AD severity [85]. CX3CL1 and its receptor CX3CR1 may regulate the activation of microglia by controlling the release of inflammatory cytokines and synaptic plasticity and cognitive functions by modulating receptors in neurons directly or indirectly. The involvement of CX3CL1/CX3CR1 in AD suggests that CX3CL1/CX3CR1 contributes positively to neuron protective as well as to the course of the disease [86]. Even though the role of LCN2 is still debated, Kang and colleagues demonstrated that LCN2 acts as an anti-inflammatory agent modulating both the peripheral and the CNS responses during systemic LPS-induced inflammation. The same authors also noticed increasing proteomic and transcriptomic levels of cytokines and chemokines, in LPS-treated Lcn2−/− relative to LPS treated WT mice [73]. This suggests that LCN2 also mitigates CNS chemokine and cytokines responses that indeed resulted to be poorly expressed in this cluster as well as almost the other neuroinflammatory factors. Therefore, the faint inflammatory reaction observed in AD-CL3 cluster could explain the slower disease progression that characterizes this group and leaves the door open to possible interventions aimed at mitigating the severity of the disease.

In conclusion, our study (i) indicates that microglial cells undergo morphological and functional changes in AD brains which most probably overtake the M1/M2 dichotomy; (ii) reveals an unprecedented association between AD and CXCL13; (iii) suggests that neuroinflammation mediated by microglia may affect several neuropathological and biochemical features of AD and may modulate the clinical/neuropathological picture of the disease; (iv) provides a potential basis to stratify patients according to their neuroinflammatory profile.

The low number of cases available for this work limited the possibility of studying in depth the correlations between clinical/neuropathological variability of AD patients and the different patterns of microglial activation. We are aware that larger cohorts of patients and controls are needed to dissect the correlations among biological, clinical and inflammatory data in AD and to understand the specific value of any molecular pathways in the generation of divergent AD phenotypes.

## 4. Materials and Methods

### 4.1. Sample Collection

All procedures for sample collection and experimental studies were in accordance with the 1964 Declaration of Helsinki and its later amendments and were approved by the Ethical Committee of “Fondazione IRCCS Istituto Neurologico C. Besta” (Milan, Italy) (REFRAME JPND 2015 project, code JPCOFUND_FP-829-085, Ethical Committee approval date: 9 September 2016). For this study we used the same cohort of patients previously analyzed and described in our lab [14]. Briefly, 24 AD brains and 6 non-demented age-matched controls were employed (Appendix A). Within the AD group, 20 cases were sporadic and 4 cases were familiar associated with the following mutations: APPA673V (fAD1), APPA713T (fAD2), PS1P117A (fAD3), and PS2A85V (fAD4). Autoptic brain samples from the AD patients and non-demented subjects were obtained at autopsy at the “Fondazione IRCCS Istituto Neurologico C. Besta” (Milan, Italy) and the Indiana University School of Medicine (Indianapolis, IN, USA).

### 4.2. Neuropathological Assessment

The neuropathological assessment was carried out according to the international guidelines for the neuropathological diagnosis of AD [87], as previously described [14]. Samples of frontal cortex were fixed in 4% formalin, dehydrated in graded ethanol (EtOH) (from 70% to 100%), cleared in xylene, embedded in paraffin, sectioned (4μ-thick) using a Semi-automatic precision CUT 5062 microtome (SLEE Medical GmbH, Mainz, Germany) and incubated at 37 °C overnight. Sections set at 56 °C for 12 min were gradually hydrated in EtOH scalar concentrations. The routine examination was carried out on sections stained with hematoxylin–eosin (H&E) and cresyl violet for Nissl substance.

### 4.3. Immunohistochemistry

Immunohistochemistry (IHC) with anti-Ionized calcium-binding adaptor protein-1 (IBA1) antibody (Wako 019-19741) (FUJIFILM Wako Chemicals U.S.A., Richmond, VA, United States) was performed to visualize both the activated and the resting phenotype of microglia. Formalin-fixed frontal cortex samples from the AD group (*n* = 15) were stained to disclose microglial cells within the brain parenchyma. Sections were heated for antigen retrieval in a Tris-EDTA solution at pH 9 by means the Pt-link instrument (Dako, Agilent Technologies, Santa Clara, CA, USA) according to the instrument manufactories. Slices were incubated 1 h at room temperature (RT) in rabbit polyclonal anti-IBA1 antibody (1:400). The immunoreactions were visualized by the EnVision Plus/Horseradish Peroxidase system for rabbit immunoglobulins (1:400) (Dako, Agilent Technologies, Santa Clara, CA, USA) using 3-3′-diaminobenzidine as chromogen (DakoCytomation, Agilent Technologies, Santa Clara, CA, USA). IHC was also performed for recognizing Aβ deposits. Before Aβ immunostaining, the sections were pre-treated with formic acid (80%, 60 min). The anti-Aβ antibodies used for this purpose were as follows: a monoclonal antibody reactive to amino acid residues 17–24 of Aβ (clone 4G8, 1:2000, BioLegend, San Diego, CA, USA) and monoclonal antibodies reactive to the C-terminus of Aβ, specifically recognizing Aβ40 (clone 11A50-B10, 1:1000, BioLegend, San Diego, CA, USA), and Aβ42 (clone 12F4, 1:500, BioLegend, San Diego, CA, USA). All sections were counterstained with Hematoxylin before being cover-slipped. The omission of the primary or the secondary antibody resulted in the absence of immunostaining (data not shown).

### 4.4. Image Analysis

Digital images were acquired using the Aperio Scanscope XT (Leica microsystems, Wetzlar, Germany) with a resolution of 40×. Images were then analyzed with the ImageScope software (Leica microsystems, Wetzlar, Germany) as follows. Ten regions of interest (ROI) drawn as squares of approximate 1 million μm^2^ were randomly designed into the grey matter layers (Appendix A). The Positive Pixel Count V9 algorithm was selected to analyze the number and the intensity of strong positive pixels. Input parameters are shown in Appendix A. The morphometric analysis was carried out by ImageJ software [88]. To determinate the plaques heterogeneity, we calculated the DI following a protocol previously described [89]. Briefly, the immunostained sections (4G8) were acquired under the light microscope Nikon Eclipse E800 (Nikon, Minato, Tokyo, Japan) by the digital camera Nikon DXM 1200 and analyzed with NIS Elements software (Nikon, Minato, Tokyo, Japan). Images of each ROI were captured (4×) and manually adjusted to erase signal background and artefacts. The total ‘‘amyloid burden’’ was defined as the mean of objects count in standard ROI: over 3 measures (object count) selected according to random sampling. The area fraction, expressed in percent by software as the total area occupied by counted objects, has been used as the denominator to calculate the DI according to the following equation:(1)$$\mathrm{DI}=\frac{1}{R}\overset{R}{\underset{r=1}{\sum }}\frac{\mathrm{object}\;\mathrm{count}\;\mathrm{of}\;r-th\;\mathrm{ROI}\;}{\mathrm{Area}\;\mathrm{fraction}\;\mathrm{of}\;r-th\;\mathrm{ROI}}$$
where R is the number of considered ROIs.

### 4.5. Preparation of Brain Homogenates and Measurement of the Levels of Neuroinflammatory Cytokines

Luminex Human Magnetic Assay (25-Plex) LXSAHM-25 (Bio-Techne R&Dsystem, Minneapolis, MN, USA) was used to quantify the abundance of 25 molecular factors (Appendix A), including cytokines, chemokines, MMPs, and IIFs released during the neuroinflammatory process by microglia. Brain samples were suspended in 9 volumes of PBS 1× and manually homogenized by a potter in ice. Brain homogenates were centrifuged for 15′ at 1500× *g* at 4 °C and the supernatant was collected and stored at −20 °C. On the day of assays, 50 µL aliquots of standard or samples were added in duplicate to the 96-well plate provide by the LXSAHM-25 assay kit (Bio-Techne R&Dsystem, Minneapolis, MN, USA). All reagents were prepared according to manufacturer instructions without modifications. Plates were read by Bio-Plex 200 System with Bio-Plex Manager™ 6.0 software (Bio-rad, Hercules, CA, USA). The Spike/Recovery test and the Linearity test were performed to assess any matrix interference and the right dilution for our samples in order to validate the use of the assay to analyze brain homogenates. Results indicated no matrix interference, therefore samples were loaded undiluted. For those analytes out or over of range (<OOR and >OOR), the 90% Minimum Detection Dose and 125% of the Maximum detection dose were used for individual samples and cytokines respectively [40,90].

### 4.6. Measurement of Aβ Levels in Brain Samples

To measure Aβ levels in brain tissue of AD patients, 200 mg of frontal cortex were homogenized in 7 volumes of 20 mM Tris-HCl, pH 7.5 added with Complete Protease Inhibitor Cocktail (Roche, Basel, Switzerland), using a manual Dounce homogenizer and ultracentrifuged at 100,000× *g* for 1 h at 4 °C. The supernatant was collected (Soluble Fraction). The pellet was re-homogenized in 20 mM Tris-HCl, 2% SDS, pH 7.5 added with Complete Protease Inhibitor Cocktail, and ultracentrifuged at 100,000× *g* for 1 h at 4 °C. The supernatant was saved and the pellet was extracted with 70% Formic Acid, sonicated for 1 min using an ultrasonic homogenizer (Sonopuls) (Bandelin electronic, Berlin, Germany), and neutralized with 20 volumes 1M Tris (Insoluble Fraction). Aβ40 and Aβ42 levels were measured in soluble (Aβ40s and Aβ42s) and insoluble (Aβ40i and Aβ42i) fractions of brain homogenates by ELISA (Millipore, Burlington, MA, USA), following the manufacturer guidelines.

### 4.7. Statistical Analysis

Statistical analysis was carried out using the R software version 3.5.1 (R Core Team 2017, Foundation for Statistical Compusting, Vienna, Austria) [91], with the significance level set at *p* < 0.05. Results obtained from the multiplex assay were previously tested using the Kolmogorov–Smirnov test and the Shapiro–Wilk test in order to assess the normal distribution of data. A non-normal distribution was found for all the analytes tested—except for IL-13—therefore, the non-parametric Wilcoxon test was chosen to compare data. Observations were considered outlier outside 1.5 times the interquartile range above the upper quartile and below the lower quartile of the boxplot. The Search Tool for the Retrieval of Interacting Genes/Proteins database (STRING v10.5) [92] was used to construct the PPI network associated with our given list of neuroinflammatory factors as input. All the direct (physical) and indirect (functional) interactions between them were derived from high-throughput lab experiments and curated databases at a high level of confidence (≥0.90). We grouped the analytes into the four biological classes to which they belong (cytokines, chemokines, MMPs, and IIFs), and data were normalized according to the relative percentage of each group (with respect to the total analytes). Then we tested the relative abundance in each group per single patient dividing each row of the data matrix (all the analytes measured per patients) by its sum (i.e., the total amount of analytes expressed per patient), obtaining a compositional dataset [93]. This compositional dataset was objective for the HCA based on the euclidean distance and the complete linkage [94]. The resulting clusters have been characterized by inspecting stratified means and standard deviations of clinical and biochemical variables.

## Figures and Tables

**Figure 1 ijms-22-02780-f001:**
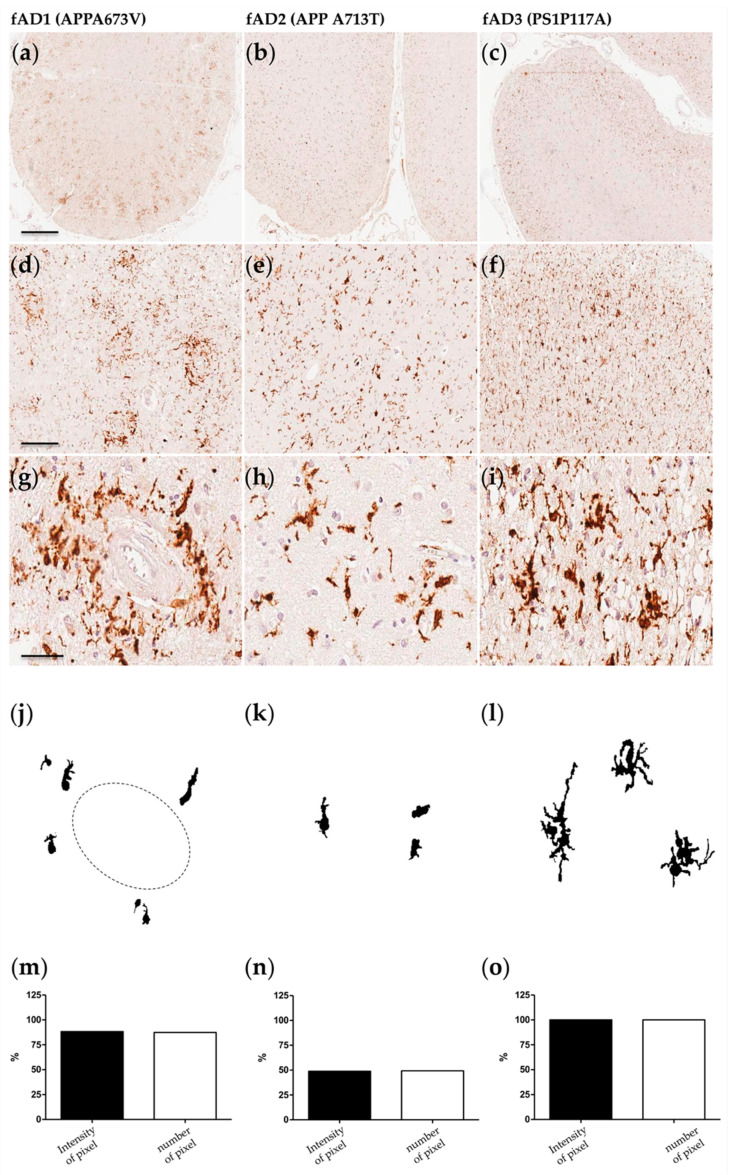
Microglial characterization in fAD cases. (**a**,**d**,**g**,**j**,**m**) APPA673V (fAD1); (**b**,**e**,**h**,**k**,**n**) APPA713V (fAD2); (**c**,**f**,**i**,**l**,**o**) PS1P117A (fAD3). Scale bar 1 mm (**a**–**c**) 200 μm (**d**–**f**); 50 μm (**g**–**i**). (**a**–**i**) Frontal cortex sections immunostained with the IBA1 antibody. (**j**–**l**) Morphological shape extracted and edited by imageJ software analysis. (**m**–**o**) Quantification of IBA1 immunoreactivity based on the intensity and number of pixels (data were normalized respect to the higher signal obtained among Alzheimer’s disease (AD) samples).

**Figure 2 ijms-22-02780-f002:**
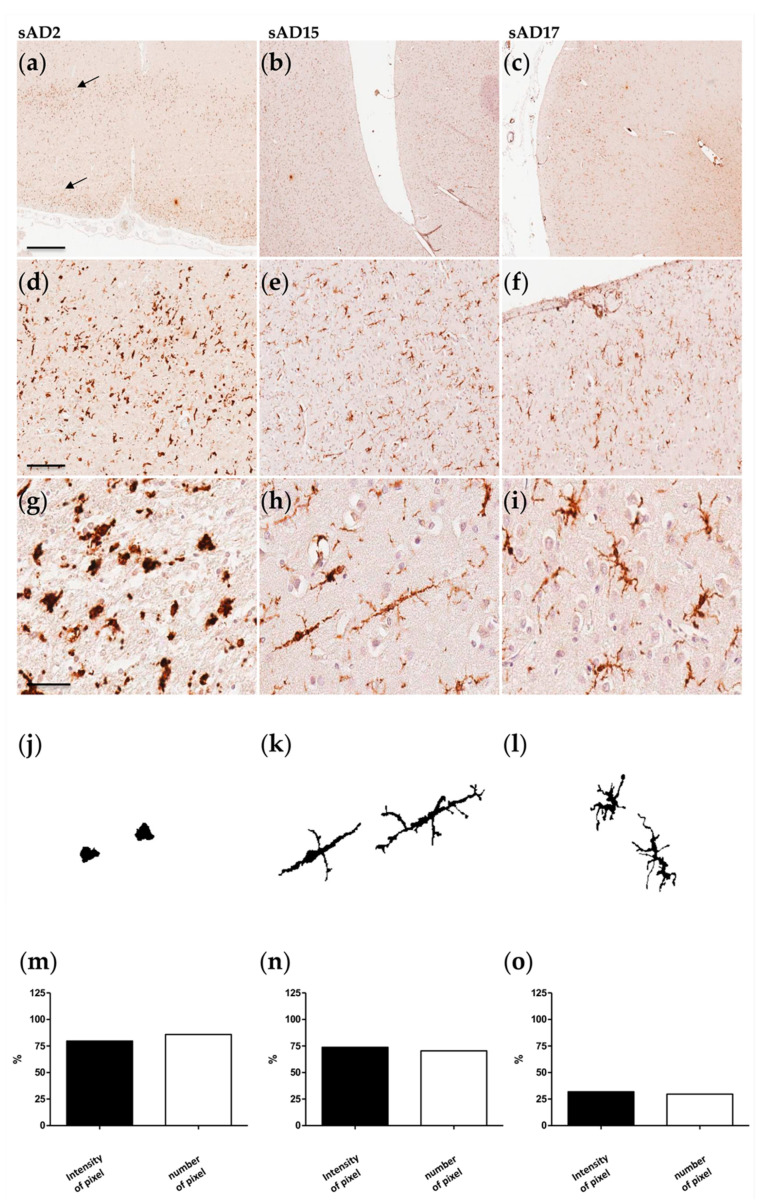
Microglial characterization in sAD cases. (**a**,**d**,**g**,**j**,**m**) sAD6; (**b**,**e**,**h**,**k**,**n**) sAD19; (**c**,**f**,**i**,**l**,**o**) sAD21. Scale bar 1 mm (**a**–**c**) 200 μm (**d**–**f**); 50 μm (**g**–**i**). (**a**–**i**) Frontal cortex sections immunostained with the IBA1 antibody. Arrows in (a) highlight the bilayer distribution of microglial cells across the frontal cortex. (**j**–**l**) Morphological shape extracted and edited by imageJ software analysis. (**m**–**o**) Quantification of IBA1 immunoreactivity based on the intensity and number of pixels (data were normalized respect to the higher signal obtained among AD samples).

**Figure 3 ijms-22-02780-f003:**
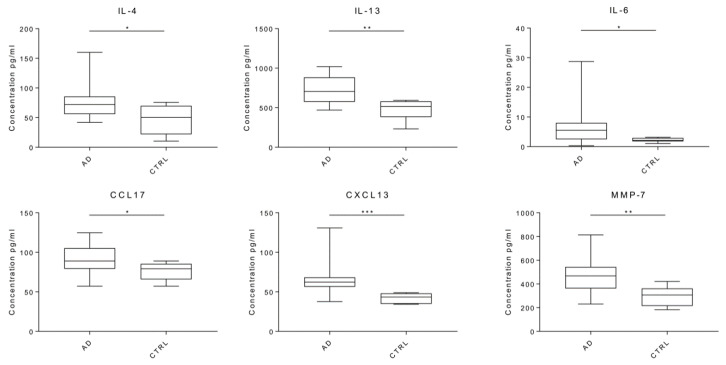
Comparison between AD and control samples. Boxplots of the most significant analytes, IL-4, IL-6, IL-13, CCL17, MMP-7, CXCL13. * *p* < 0.05, ** *p* < 0.005, *** *p* < 0.001. For IL-6 the most extreme observations have been omitted for a better graphical representation.

**Figure 4 ijms-22-02780-f004:**
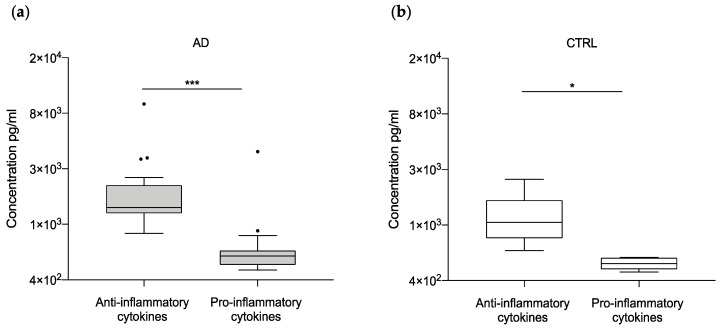
Comparison between pro- and anti-inflammatory cytokines in AD and control samples. Boxplot of the pro-inflammatory (IFN-γ, IL-1ɑ, IL-2, IL-6, IL-12 p70, IL-18) and the anti-inflammatory (IL-1rn, IL-4, IL-13) cytokines are shown for AD (*p*-value < 0.0001) (**a**) and control (*p*-value 0.29) (**b**) group. * *p* < 0.05, *** *p* < 0.001. Dots indicate outlier values. Natural logarithm scale has been used for a better graphical representation.

**Figure 5 ijms-22-02780-f005:**
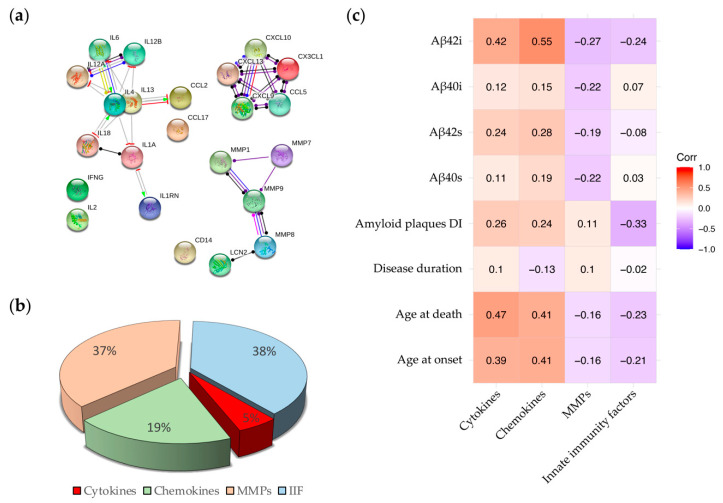
Families of neuroinflammatory factors. (**a**) STRING network highlights the direct (physical) and indirect (functional) connection of elements belongs to each family. Known interactions: from curated databases (light blue), experimentally determined (purple); predicted interactions: gene neighborhood (green), gene fusions (red), gene co-occurrence (blue); other interactions: text mining (yellow), co-expression (black), protein homology (grey). (**b**) Pie graph indicates the relative distribution of cytokines (red), chemokines (green), MMPs (pink) and IIFs (blue) within the AD group. (**c**) The heat map shows correlations between the four neuroinflammatory classes and clinical, neuropathological and biological data. Aβ42 levels in the insoluble fraction of brain homogenates (Aβ42i); Aβ40 levels in the insoluble fraction of brain homogenates (Aβ40i); Aβ42 levels in the soluble fraction of brain homogenates (Aβ42s); Aβ40 levels in the soluble fraction of brain homogenates (Aβ40s).

**Figure 6 ijms-22-02780-f006:**
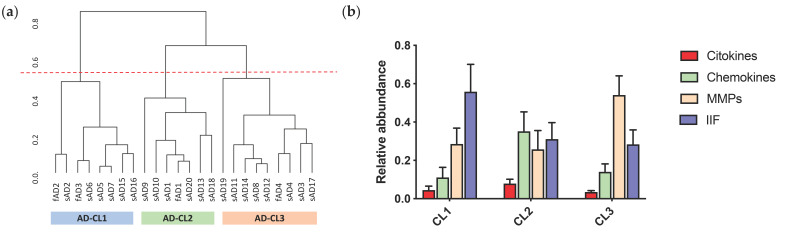
(**a**) Hierarchical Cluster Analysis (HCA) shows the different subgroups of AD patients characterized by different expression of neuroinflammatory molecular factors. In abscissa, there are the AD patients and the ordinate corresponds to the complete linkage measured by Euclidean distance. The dashed red line represents the cut off for three clusters (CL): AD-CL1 in yellow; AD-CL2 in red; AD-CL3 in light blue. (**b**) Histogram concerning the relative abundance of each inflammatory family (cytokines in red, chemokines in green, matrix-metalloproteinases (MMPs) in cream and Innate immunity factors in blue) within each cluster.

**Table 1 ijms-22-02780-t001:** The median (μ1/2) and the rate of measurements (pg/mL) for the detectable analytes in AD versus control. Values in bold underline the statistically significant differences between the two groups of samples. * *p* < 0.05, ** *p* < 0.005, *** *p* < 0.001.

Analytes	Cases μ1/2	Controls μ1/2	Rate	*p*-Values	
IL-1rn	744.120	707.42	1.051	0.4316	
**IL-4**	**72.155**	**50.18**	**1.578**	**0.0357**	*****
**IL-13**	**706.18**	**514.71**	**1.372**	**0.0050**	******
IFN-γ	51.54	48.13	1.071	0.5682	
IL-1ɑ	9.67	8.67	1.115	0.1261	
IL-2	332.09	296.64	1.119	0.0584	
**IL-6**	**5.85**	**2.11**	**2.770**	**0.0183**	*****
IL-12 p70	134.48	124.08	1.083	0.4364	
IL-18	66.68	62.27	1.071	0.0863	
CCL2/MCP1	103.29	54.43	1.898	0.0971	
CCL5/RANTES	29.19	15.27	1.912	0.0518	
**CCL17/TARC**	**89.11**	**79.17**	**1.126**	**0.0447**	*****
CX3CL1/Fractalkine	6793.00	6225.31	1.091	0.4622	
CXCL9/MIG	869.11	493.92	1.760	0.1538	
CXCL10/IP-10	10.98	6.49	1.693	0.1156	
**CXCL13/BLC/BCA-1**	**62.37**	**43.41**	**1.437**	**0.0008**	*******
MMP-1	226.17	181.96	1.243	0.0584	
**MMP-7**	**468.31**	**306.91**	**1.526**	**0.0041**	******
MMP-8	5396.34	3516.87	1.534	0.1740	
MMP-9	12,305.59	11,272.82	1.092	0.7047	
CD14	10,364.04	8948.05	1.158	0.2099	
Lipocalin-2/NGAL	5585.46	4475.46	1.248	0.1286	

**Table 2 ijms-22-02780-t002:** Cluster characterization. The table recapitulates the average for all the clinical, neuropathological and biological data available for each cluster. ±: standard deviation. DI: Dispersion Index, CAA: Cerebral Amyloid Angiopathy, Aβ40s: Aβ40 soluble levels, Aβ42s: Aβ42 soluble levels, Aβ40i: Aβ40 insoluble levels, Aβ42i: Aβ42 insoluble levels. Number and intensity of IBA1 are referred to positive pixel (pp) measurement. *p*-value calculated with Kruskal Wallis test. *p* < 0.05 *, ns = not significative.

	AD-CL1 (*n* = 8)	AD-CL2 (*n* = 7)	AD-CL3 (*n* = 9)	*p*-Value
Age at onset	56.13 (±12.06)	67.15 (±18.48)	60.88 (±9.21)	ns
Age at death	63.38 (±12.95)	73.86 (±15.23)	67.3 (±9.06)	ns
Disease duration	7.25 (±3.60)	6.71 (±3.84)	13.2 (±0.10)	ns
Brak stage	5.88 (±0.22)	4.86 (±1.12)	5.22 (±0.97)	ns
DI	144.01 (±62.67)	175.82 (±94.83)	159.76 (±59.72)	ns
CAA	2.64 (±0.87)	2.86 (±0.83)	2.00 (±0.92)	ns
IBA1 pp intensity	1.93 × 10^8^ (±1.19 × 10^8^)	1.59 × 10^8^ (±1.07 × 10^8^)	1.01 ×10^8^ (±8.27 × 10^8^)	ns
IBA1 pp number	2.66 × 10^6^ (±1.65 × 10^6^)	2.07 × 10^6^ (±1.45 × 10^6^)	1.29 × 10^6^ (±1.07 × 10^6^)	ns
Aβ40s	39.191 (±56.72)	273.26 (±428.71)	10.92 (±10.26)	ns
Aβ42s	1.76 (±0.99)	8.22 (±11.00)	2.90 (±2.02)	*
Aβ40i	3860.80 (±10450.27)	45,565.88 (±102746.20)	111.91 (±84.84)	ns
Aβ42i	49.88 (±27.72)	102.01 (±37.34)	65.32 (±42.59)	ns

## Data Availability

The data presented in the study including the Appendix A are freely accessible. The raw data produced in the study are available on request from the corresponding author.

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
