# Peer review of "Microglial Heterogeneity and Its Potential Role in Driving Phenotypic Diversity of Alzheimer’s Disease"

_ijms, 2021, doi:10.3390/ijms22052780_

Round 1

Reviewer 1 Report

In the paper "Microglial Heterogeneity and its potential role in driving phenotypic diversity of Alzheimer's disease" by Sorrentino et al., the authors want to unravel the potential contribution of microglia to the variability observed in Alzheimer's disease phenotype. This aspect is well supported by a plethora of very recent reports in the literature, describing at cellular and molecular level the presence of heterogeneous population of microglia in different brain areas and its behaviour in different pathologies. In this sense this paper add a contribution to these findings, despite the low number of cases analyzed, as also reported by the authors.

There are few points I would like to point to:

- the introduction should be improved with more updated references, for example concerning the association between neuroinflammation and the development of AD (the two mentioned are from 2013), but also concerning microglia. An example are:

> Marta Sochocka & Breno Satler Diniz & Jerzy Leszek - 2017 - Inflammatory Response in the CNS: Friend or Foe? - Mol Neurobiol (2017) 54:8071–8089

> Arsalan Hashemiaghdama, Magdalena Mroczekb - 2020 - Microglia heterogeneity and neurodegeneration: The emerging paradigm of the role of immunity in Alzheimer's disease

> Srinivasan et al. - 2020 - Alzheimer’s Patient Microglia Exhibit Enhanced Aging and Unique Transcriptional Activation - Cell Reports 31, 107843

> Heela Sarlus, Michael T. Heneka - 2017 - Microglia in Alzheimer’s disease. J Clin Invest. 2017;127(9):3240-3249

> Stratoulias et al. - 2019 - Microglial subtypes: diversity within the microglial community. The EMBO Journal 38: e101997

- the discussion may also be improved (lane 351, lanes 358, 359) by citating and discussing papers as:

> Hajii et al. - 2019 - Ultrastructural evidence of microglial heterogeneity in Alzheimer’s disease amyloid pathology. Journal of Neuroinflammation (2019) 16:87

> Böttcher et al. - 2019 - Human microglia regional heterogeneity and phenotypes determined by multiplexed single-cell mass cytometry. Nature Neuroscience 22: 78–90.

- In the experiments described in paragraph 2.3, the authors looked for a correlation between inflammatory molecules and clinical, biochemical and neuropathological features of AD, by grouping molecules into four classes (Figure 5). I understand that this may facilitate the analysis and a possible stratification of patients but there is the limit of missing some important data. Concerning figure 5c, they ended up (lane 266) saying that the correlation between MMPs and all the disease indicators are weak and for this reason they cannot give any conclusion. MMPs in figure 5b are one of the two most represented categories and it is quite unaspected not to find any correlation. Indeed, in figure S3 two out of four MMPs genes (MMP1 and MMP8) show a stronger correlation (both positive and negative) with some disease indicators. Probably the authors should discuss better this point and consider to divide these molecules into two subclasses. Grouping them together may hide some important information.

- minor points are:

lane 221. in figure 5a CXCL1 (in the text) is indicated as CX3CL1, whereas CX3CL9 (in the text) is indicated as CXCL9. Please fix it.

in the legend of figure 5c, lanes 240-241, Abeta42s and Abeta40s are indicated again as "insoluble", but they should be "soluble".

Author Response

We would like to thank the reviewers as well as the editor for the effort they have expended reviewing this manuscript. We have carefully considered each point and have made changes that we hope have substantially improved the quality of the manuscript. Changes applied to the manuscript have been highlighted by "Track Changes" function in the Microsoft Word document as suggested by the editor.

Author responses to the comments of reviewers are highlighted in red and have been addressed point by point below.

Reviewer #1

- the introduction should be improved with more updated references, for example concerning the association between neuroinflammation and the development of AD (the two mentioned are from 2013), but also concerning microglia. An example are:

> Marta Sochocka & Breno Satler Diniz & Jerzy Leszek - 2017 - Inflammatory Response in the CNS: Friend or Foe? - Mol Neurobiol (2017) 54:8071–8089

In response to the comment of the reviewer, we substituted the previous references, (i.e. “17-18”, lane 68), with Hashemiaghdama et. al., 2020 and Srinivasan et. al., 2020.

We are familiar with the work of Sochocka et al and we already cited this study (ref 61) among the references in the discussion paragraph, lane 417.

> Arsalan Hashemiaghdama, Magdalena Mroczekb - 2020 - Microglia heterogeneity and neurodegeneration: The emerging paradigm of the role of immunity in Alzheimer's disease

We added the above reference on lane 68. It is now listed in the reference paragraph with the number 18.

> Srinivasan et al. - 2020 - Alzheimer’s Patient Microglia Exhibit Enhanced Aging and Unique Transcriptional Activation - Cell Reports 31, 107843

We added the above reference on lane 68. It is now listed in the reference paragraph with the number 17.

> Heela Sarlus, Michael T. Heneka - 2017 - Microglia in Alzheimer’s disease. J Clin Invest. 2017;127(9):3240-3249

We cited the above reference on lane 46. It is now listed in the reference paragraph with the number 5.

> Stratoulias et al. - 2019 - Microglial subtypes: diversity within the microglial community. The EMBO Journal 38: e101997

We cited the above reference on lane 71. It is now listed in the reference paragraph with the number 19.

- the discussion may also be improved (lane 351, lanes 358, 359) by citating and discussing papers as:

> Haji et al. - 2019 - Ultrastructural evidence of microglial heterogeneity in Alzheimer’s disease amyloid pathology. Journal of Neuroinflammation (2019) 16:87

> Böttcher et al. - 2019 - Human microglia regional heterogeneity and phenotypes determined by multiplexed single-cell mass cytometry. Nature Neuroscience 22: 78–90.

We appreciated the suggestion of the reviewer and carefully read the articles cited by the referee to discuss commonalities and diversities with respect to their current work. The discussion paragraph has been added with extra text and references at line 365-373 and 389-396.

Haji et al. - 2019 - Ultrastructural evidence of microglial heterogeneity in Alzheimer’s disease amyloid pathology. Journal of Neuroinflammation (2019) 16:87

This article cited by the referee reinforces, in our opinion, the heterogeneity of microglial cells in the pathological context of AD supporting the observations reported in our work. Despite the different level of analysis, we found morphological parallels with the features of microglia described by Haji et al., in the fibrillar amyloid-β areas. The phagocytic microglia with enlarged cell body - immunoreactive for IBA1 in both processes and cell bodies - described by the authors at ultrastructure level was also observed in microglia associated plaques in our patients.

We explained this point in the “discussion” section from lane 365 to lane 373:

“Such morphological differences were also highlighted by the study of El Hajj et al., from an ultrastructural point of view [53]. By dividing hippocampal areas of APPSwe-PS1Δe9 mice into subregions, according to pathological insult, the authors identified distinct microglial ultrastructural aberrations related to AD pathology. Importantly, several of these ultrastructural alterations were also observed in the human hippocampus of AD patients. In particular, phagocytic microglia recognized in fibrillar amyloid-β areas look like, although at pure morphological level, microglia associated plaques observed in this paper with enlarged cell bodies in the proximity of Aβ deposits.”

> Böttcher et al. - 2019 - Human microglia regional heterogeneity and phenotypes determined by multiplexed single-cell mass cytometry. Nature Neuroscience 22: 78–90.

This article well addresses the problem of microglial regional heterogeneity demonstrating - at the single-cell level - the biological individuality of microglia (in respect to other near cell lineages) and importantly, identifies specific microglial subsets according to the regional brain allocation. So, we would like to thank the reviewer to put under our attention this work. Although Böttcher et al., do not focus on a specific pathology, we believe their work could be further useful to explain the regional vulnerability observed in several neurodegenerative disorders, including AD. Our work did not take into account different brain areas but reported microglial variability within the same region as well, and focused on the analysis of secreted molecules, which can be potentially measured in biofluids, like CSF or blood, for further biomarker studies. Anyway, we think that both studies strongly support the role of microglial heterogeneity in modulating AD phenotypes.

We discussed this point by adding the following text in the “discussion” section from lane 389 to lane 396:

“Recently, through the use of multiplexed mass cytometry, Böttcher and colleagues have clearly demonstrated the presence of specific microglial subsets, each one characterized by a peculiar signature, examining distinct human brain areas [57]. The existence of human microglial phenotypes in neurodegenerative diseases supports our attempt to demonstrate the importance of the microglial heterogeneity in the pathological context of AD. Indeed, each microglial subset could reasonably impact on the neuroinflammatory response differently, leading to selective vulnerability to AD pathology in distinct brain areas and being a source of phenotypic variability in AD patients”.

- In the experiments described in paragraph 2.3, the authors looked for a correlation between inflammatory molecules and clinical, biochemical and neuropathological features of AD, by grouping molecules into four classes (Figure 5). I understand that this may facilitate the analysis and a possible stratification of patients but there is the limit of missing some important data. Concerning figure 5c, they ended up (lane 266) saying that the correlation between MMPs and all the disease indicators are weak and for this reason they cannot give any conclusion. MMPs in figure 5b are one of the two most represented categories and it is quite unaspected not to find any correlation. Indeed, in figure S3 two out of four MMPs genes (MMP1 and MMP8) show a stronger correlation (both positive and negative) with some disease indicators. Probably the authors should discuss better this point and consider to divide these molecules into two subclasses. Grouping them together may hide some important information.

The authors appreciated this comment and agree with the reviewer on the grouping effect. Following the advice of the reviewer, we tried to subdivide the MMP family into 2 subgroups according to their functional classification. We added this result as supplementary figure S6. In this way, MMP-1 and MMP-8 would be part of the same group (i.e., collagenases). However, MMP-1 is poorly represented in our cohort so, when we grouped it with MMP-8, we obtained a subgroup highly correlated with MMP-8 (i.e., Pearson's correlation coefficient among MMP-8 and the new subgroup is equal to 0.999). This means that grouping MMP-1 and MMP-8 together leads to masking the positive association between MMP-1 and disease indicators. For these reasons, we believe it could be plausible keeping the current grouping of MMPs and describe the contribution of each one in the main text. To address this point, we explained more in details the MMPs behavior in the result subsection (lanes 271-277) and in the discussion paragraph (lanes 446-457). Moreover, considering the peculiar findings regarding the MMP-8, we further discuss (lanes 489-495) its correlation with the AD-CL3 and its possible association to cerebrovascular dysfunctions in AD-CL3 cluster. To make easier the comprehension of these findings and the discussion of results, we included the figure below in the Supplementary Material as Figure S6.

- minor points are:

lane 221. in figure 5a CXCL1 (in the text) is indicated as CX3CL1, whereas CX3CL9 (in the text) is indicated as CXCL9. Please fix it.

This was adjusted in the main text

in the legend of figure 5c, lanes 240-241, Abeta42s and Abeta40s are indicated again as "insoluble", but they should be "soluble".

This was corrected in the legend of figure 5c

Reviewer 2 Report

Authors presented the possible effects of inflammation cytokines from glial cells and their role in AD in diverse AD patients, especially in fAD brains.

Authors noticed the different patterns of changes in cytokines in AD patients and grouped them.

This is very interesting and significant report, as well as well written at present form.

Author Response

We would like to thank the reviewer for his/her positive comments on our study.